# Evolution of Telehealth—Its Impact on Palliative Care and Medication Management

**DOI:** 10.3390/pharmacy12020061

**Published:** 2024-04-02

**Authors:** Syed N. Imam, Ursula K. Braun, Mary A. Garcia, Leanne K. Jackson

**Affiliations:** 1Office of Connected Care, Michael E. DeBakey Veteran Affairs Medical Center, Houston, TX 77030, USA; 2Department of Medicine, Section of Geriatric and Palliative Medicine, Baylor College of Medicine, Houston, TX 77030, USA; 3Rehabilitation & Extended Care Line, Section of Palliative Medicine, Michael E. DeBakey Veteran Affairs Medical Center, Houston, TX 77030, USA

**Keywords:** telehealth, telemedicine, history of telemedicine, palliative care, medication management, benefits of telehealth, limitation of telehealth, telepharmacy

## Abstract

Palliative care plays a crucial role in enhancing the quality of life for individuals facing serious illnesses, aiming to alleviate suffering and provide holistic support. With the advent of telehealth, there is a growing interest in leveraging technology to extend the reach and effectiveness of palliative care services. This article provides a comprehensive review of the evolution of telehealth, the current state of telemedicine in palliative care, and the role of telepharmacy and medication management. Herein we highlight the potential benefits, challenges, and future directions of palliative telemedicine. As the field continues to advance, the article proposes key considerations for future research, policy development, and clinical implementation, aiming to maximize the advantages of telehealth in assisting individuals and their families throughout the palliative care journey. The comprehensive analysis presented herein contributes to a deeper understanding of the role of telehealth in palliative care and serves as a guide for shaping its future trajectory.

## 1. Case

Mr. Alex, a 69-year-old male, was recently diagnosed with advanced pancreatic cancer and end-stage chronic obstructive pulmonary disease (COPD) on home oxygen. He was admitted to the hospital with severe abdominal pain, nausea, diarrhea, dyspnea, and significant weight loss. During his stay, Mr. Alex was seen by the multidisciplinary palliative care team to discuss prognosis, symptom management, and goals of care, including meeting emotional needs, to ensure a dignified end-of-life experience. 

Mr. Alex faced challenges with his care due to limited mobility caused by his symptoms and travel difficulties; as he lived in a rural area, making long drives was strenuous for him. Due to his high degree of symptom burden and frequent need for clinic visits to assess and titrate medication use, the palliative care team offered to integrate telehealth into Mr. Alex’s care plan. Regular virtual consultations were conducted through a secure telehealth platform, allowing healthcare providers to assess and address Mr. Alex’s evolving needs remotely. Symptom management, emotional support, and family counseling were all managed virtually. 

Synchronous virtual consultations enabled timely assessment and adjustment of Mr. Alex’s pain, nausea, and diarrhea management regimen. Mr. Alex could report symptoms and side effects from the comfort of his home, allowing for prompt intervention and optimization of medications. Telemedicine allowed this medically vulnerable patient to limit unnecessary exposure to other sick patients and allowed him to conserve his home oxygen tanks by obviating the need to travel to the clinic.

Telehealth facilitated regular virtual sessions with a palliative care psychologist, offering emotional support to both the patient and his family, addressing anxiety, depression, and concerns related to end-of-life care. Family members participated in virtual care planning meetings, ensuring their active involvement in decision-making processes. Telehealth allowed for open communication, education on caregiving, and the provision of resources to support the family in their caregiving role. Through synchronous video consultation, Mr. Alex’s quality of life improved significantly. Receiving care at home minimized the stress associated with travel, allowing him to focus on spending meaningful time with loved ones. 

## 2. Article Scope and Intention

The aim of this general review is to analyze themes that have emerged to identify commonalities and gaps in the current literature. On 11 March 2024, the reviewers accessed the Web of Science Core Collection using the topic search terms “telemed* or telehealth* or telepall* or telehospice*” AND “palliative* or supportive care or supportive medicine”. The search yielded 979 results, the vast majority of which have been published since 2020 (Figure 1). Using the same key words in Pubmed, accessed on 11 March 2024, with the search limited to “Title/Abstract”, yielded 549 articles published since 1986, again with the majority being from the past 4 years, suggesting it is a rapidly evolving field (Figure 2). 

This broad overview includes a wide range of topics and perspectives within the telehealth domain, drawing on the experience of telemedicine within management of chronic health conditions such as type 2 diabetes, dyslipidemia, hypertension/cardiovascular medicine, depression, and cancer pain and care during the COVID-19 pandemic, as well as within the field of palliative care. We explore various models of care and geographic areas impacted by telehealth and how telemedicine impacts patients, caregivers, medical professionals, and healthcare systems, including its potential benefits, challenges, and future directions. 

## 3. Background

### Telehealth Evolution

Pioneering applications of technological inventions in healthcare and medicine have shaped patient care. But, despite the availability and known benefits of the telemedicine concept and technology, it was not adopted widely until the COVID-19 pandemic. 

Pre-COVID-19 pandemic, the Medicare insurance reimbursement structure was one of the major barriers to the growth of virtual visits, despite the available technology and patient interest. The unprecedented circumstances due to the COVID-19 pandemic and changes in the reimbursement structure resulted in an exponential growth of telemedicine. 

The Centers for Medicare and Medicaid Services (CMS) issued an expansion of telehealth with the 1135 waiver in March 2020 for the duration of the COVID-19 pandemic national emergency [1]. The 1135 waiver allowed CMS to waive certain requirements during the COVID-19 national emergency, allowing United States physicians to provide telemedicine across state lines and reimbursement in both inpatient and outpatient settings. Before the waiver, Medicare payment was restricted to specific telehealth visits depending upon the patient’s location and type of service provided. 

The first spoken words transmitted electronically by telephone for medical assistance were those of the inventor Graham Bell on 10 March 1876 [2]. Just three years after the invention of the telephone, an article in the November 1879 edition of *The Lancet* cited a case of an anxious mother who, convinced that her baby had croup, called her family doctor at midnight seeking a home visit to evaluate her baby. The doctor asked to hear the baby’s cough on the telephone and was able to determine based on this phone assessment that it was not croup, thus avoiding an unnecessary midnight house call as well as relieving the anxiety of the baby’s family [2].

In 1925, publisher Hugo Gernsback predicted the use of telemedicine. He published a description of a device that doctors would use in the future and called it Teledactyl (Figure 3). This device would allow the remote examination of a patient using robotic fingers and a screen [3,4,5].

E-health is the umbrella term that is defined as the use of information, communication, and technologies for health. It includes both telehealth and telemedicine. While telehealth and telemedicine are often used interchangeably, telemedicine is a subset of telehealth, and telehealth is a subset of e-health. 

The Health Resources & Services Administration (HRSA) defines telehealth as the use of electronic information and telecommunications technologies to support and promote long-distance clinical healthcare, patient and professional health-related education, and public health and health administration. Technologies include videoconferencing, the internet, store and forward imaging, streaming media, and landline and wireless communications [6].

According to the American Telemedicine Association, telemedicine refers to the exchange of medical information from one site to another through electronic communications to improve a patient’s health. 

Telemedicine is an efficient method of healthcare delivery that has evolved exponentially and is now an integral part of the healthcare infrastructure. 

Synchronous telehealth is the most well-known type of technology-based healthcare service. It requires a live two-way audiovisual link between a patient and a care provider. Synchronous telehealth includes any video call or live chat software that allows a healthcare provider to communicate with a patient in real time.

## 4. Essentials of Synchronous Telehealth

Patients at home or in any healthcare institute can receive care via synchronous telemedicine. The visit requires specific technological components, including appropriate hardware, peripherals, and software. The provider can ask patients to conduct a provider-directed self-examination and utilize peripherals to augment the physical examination. Peripherals are devices utilized to digitally collect biometric data like heart sounds, EKG, skin lesion images, and vitals and transmit them to distant locations via the internet. The telemedicine cart has peripherals not available at home, which may include a digital stethoscope, digital otoscope, dermascope, or other specific peripherals that optimize diagnostic accuracy. Telemedicine carts are systems that integrate cameras, displays, peripherals, and network access to bring remote physicians to the side of the patient. 

Patient side: For a home televisit, the patient will need a smart device with video call capability such as a smartphone, tablet, laptop, or desktop. If the patient is at a healthcare institute, a telemedicine cart may be used. 

Provider side: The provider requires a secure and Health Insurance Portability and Accountability Act (HIPAA)-compliant videoconferencing application component of a telehealth platform. The application should be web- or cloud-based to allow access from different devices and locations. If possible, the application should be compatible with different operating systems and smart devices, allowing access to a larger patient population. 

Workflow system (standalone telemedicine portal or integrated with EMR): Virtual care should be carried out in a system of care that provides the necessary nursing and ancillary staff to support a comprehensive telehealth visit. This system would optimize the healthcare delivery for the patient. Systems mimic in-office workflow, can increase patients’ virtual compliance, and improve provider’s efficiency. 

It is essential that providers confirm the physical location of the patient during the virtual care if a medical emergency occurs and emergency services need to be dispatched to the patient’s location. 

## 5. Benefits of Telehealth

Telehealth provides many positive elements to healthcare for both patients and providers in regard to convenience, cost of care, chronic disease management and preventive care, and decreasing the risk of nosocomial infections. Telehealth offers patients and their families greater flexibility and reduced time away from work. Telehealth visits are conducted in the comfort and privacy of the individual’s own home. Patients do not need to travel to the medical facility, which saves time and money by decreasing travel expenditure, childcare expenses, and other related expenses. It eliminates the distance barrier, especially for patients living in rural areas, and allows for easier access for elderly patients and patients with disabilities. More family members can participate in the visit because there is no travel required. Patients and their family members can channel more time, energy, and resources into the well-being of the patient. Easier access and convenience result in high levels of patient satisfaction. Benefits for providers using telehealth include reduced overhead expenses with a reduced need for support staff and the use of fewer exam rooms. Providers have increased availability to patients as there are fewer no-shows and cancellations, allowing for greater efficiency. 

Telehealth interventions have been shown to improve medication adherence outcomes including medication possession ration (MPR) and proportion of days covered (PDC) rates [7]. Studies in diabetic, cardiovascular, and depression management have shown that pharmacist-led telehealth improves medication adherence as compared to nurse-led telehealth [8]. 

## 6. Telehealth Impact on Palliative Care

Palliative care is specialized medical care for people living with a serious illness. The care is provided by a multidisciplinary team, ideally including a clinical pharmacist, that focuses on providing relief from the symptoms and stress of the illness with a goal to improve quality of life. 

In 1967, the founder of the modern hospice movement, Dame Cicely Saunders, founded the first modern hospice, St Christopher’s Hospice, in London. The hospice movement in the 1960s gave rise to palliative care. Even though palliative care traditionally has been seen as a discipline that is hands on rather than high tech, telehealth helps overcome many hurdles in providing effective palliative care. Twenty percent of Americans living in rural areas have lower average incomes, education, and insurance coverage, often leading to suboptimal treatment of cancer and other chronic diseases. Telemedicine allows for improved high-quality palliative care for these patients. While ninety percent of hospitals in urban areas have palliative care available, out of all rural hospitals with fifty or more beds, only 17% have palliative care access [9].

Outpatient palliative care via telemedicine can help prevent hospitalizations by establishing goals of care early, identifying resources to remain at home, specifying worrisome symptoms, and educating caregivers about safe practices. 

One study conducted from 2008 to 2011 in a tertiary cancer center in Edmonton, Canada, and published in 2013 evaluated telemedicine to improve access to a specialist multidisciplinary palliative care team for rural cancer patients [10]. Visits occurred via telemedicine with the palliative care interdisciplinary team. Per visit, the estimated time savings and cost savings were 7.96 h and CAD 192, respectively [10]. 

In 2015, a new telehealth program was launched to enhance hospice and palliative care delivery in rural communities in northwest Kansas [11]. A not-for-profit, community-based organization serving rural and frontier counties was selected to partner with KUMC for the program. A total of 123 patient-related telehealth video calls were conducted with staff members and the patient, family, and/or caregivers. Interdisciplinary team (IDT) meetings and administrative meetings were also conducted through Zoom. Program savings including all patient-related, nonpatient-related, and administrative savings resulted in an average of approximately USD 19,200/month in mileage reimbursement and travel time [11]. 

To explore strategies for increasing access to palliative care among individuals living in remote and rural communities, a review was conducted on studies that explored the use of telehealth in this population [12]. The 18 studies were selected from seven countries. Most of the studies involved oncology patients and videoconferencing or a web platform that included online software with videoconferencing. Three significant issues emerged from the review: (1) delivery of care, (2) symptom management that impacted quality of life, and (3) the satisfaction level of the patients, caregivers, and providers. Patients and their caregivers felt comfortable discussing sensitive topics over video call [13]. Telehealth was found to be effective for patient and medication monitoring, provider and specialist appointments, and palliative care consultations. Operational benefits included clinician time saved, shorter appointment wait times, and reduced no-show rates. Implementation challenges of telehealth were also found in the review. Statistical improvements in quality of life and symptom management were reported. Nearly two-thirds of the studies reported positive experiences among patients, caregivers, and providers. Initial findings suggested that both patients and providers in outpatient palliative care generally report positive experiences with telehealth [14]. More research should be conducted to confirm the viability of clinical care delivery and establish best practices for quality virtual palliative care in remote and rural areas.

Telemedicine presents an innovative solution for cancer patients facing challenges in the ability to make clinic appointments due to both disabling symptoms and the difficulties of traveling long distances. Pain is a crucial factor significantly impacting the quality of life for these individuals. A Systematic Review and Meta-Analysis of Randomized Controlled Trials investigated the efficacy of telemedicine in addressing cancer pain management [15]. The review evaluated the integration of telemedicine with traditional in-person visits. Patients treated through telemedicine exhibited lower pain severity scores and a reduction in negative impacts due to pain compared to those receiving conventional care. Telemedicine proves to be an effective tool for cancer pain management and holds the potential to positively impact the economic and organizational aspects of healthcare systems.

Pain and depression are common in people who have cancer, which leads to both physical and emotional distress. Unfortunately, often patients do not discuss these issues with their medical team, resulting in inadequate treatment, despite how prevalent the issues are and the significant impact they have on the patients. A randomized controlled trial involving 405 participants from 16 community-based oncology practices over a 12-month period looked at whether centralized telephone-based care management, along with automated monitoring of symptoms, could help improve depression and pain in cancer patients [16]. The results showed that the intervention group experienced greater improvements in both pain severity (measured by Brief Pain Inventory) and depression severity (measured by Hopkins Symptom Checklist-20) compared to the usual care group. The study concluded that centralized telecare management, combined with automated symptom monitoring, can lead to better outcomes for pain and depression in cancer patients across diverse settings including oncology practices in both urban and rural areas.

To improve access to quality palliative care, MD Anderson conducted an Extension for Community Healthcare Outcomes—Palliative Care for Africa (ECHO-PACA) [17]. This program examined the advantages of the ECHO-PACA, with a specific focus on expanding access to high-quality palliative care through telehealth. The ECHO-PACA model effectively disseminates knowledge and improves skills related to palliative care among healthcare providers in underserved regions without the need for travel. 

## 7. Telehealth Impact on Medication Management and Pharmacy

The Ryan Haight Online Pharmacy Consumer Protection Act of 2008 was passed to regulate online prescriptions of controlled substances. The act requires the practitioner to evaluate the patient in person, requires online pharmacies to have modified registration, and includes the addition of criminal offenses. During the COVID-19 pandemic, telemedicine and audio–video flexibilities were passed. The COVID-19 Telemedicine Flexibilities for Prescription of Controlled Medications allowed controlled substances to be prescribed by synchronous telemedicine without face-to-face evaluation of the patient by the prescriber. The act was granted a temporary extension on 10 May 2023, and again on 10 October 2023, which allows telemedicine flexibilities for prescribing controlled substances for new practitioner–patient telemedicine relationships until 31 December 2024.

Telemedicine makes medication review easier as the patient can show the provider all their medications, including how they are stored and organized. A national survey showed that 88% of adults in the U.S. do not have the health literacy skills to manage all the demands of the healthcare system, and 36% have limited health literacy, which leads to more negative patient outcomes, including medication errors, using fewer preventative services, and worse health status [18].

A retrospective analysis of Veterans Affairs (VA) medical records in 2022 was conducted to review data from veteran participants involved in a pilot implementation of the Telepharmacy Model of Care at the Ann Arbor Healthcare System [19]. The study revealed that 98% of participants utilized telemedicine, and each patient had a median of 18 medications. Notably, 57% of patients experienced four or more medication-related discrepancies. Utilizing the Safe Medication Algorithm for Older Adults tool, 35% were flagged for Red-Flag medications and 74% for High-Risk medications. A significant percentage of participants, including 37% with cognitive and health literacy impairments, and 45% with physical impairments, had impaired ability to self-manage medications. Recommendations for deprescribing were made in 98% of cases. The study showed how telepharmacy can effectively tackle medication management challenges and improve access to pharmaceutical services, particularly in underserved or rural areas.

The risk of medication errors is high during transition of care. Studies have estimated that, at hospital discharge, more than half of patients have a medication error, and an adverse drug event is experienced by nearly 1 in 5 patients [20]. Post-discharge follow-up is critical for checking the health status of the patient and ensuring that the proper medications are being taken by the patient. Post-discharge telemedicine visits had higher completion rates than face-to-face visits according to one study [21], decreasing the risk of medication error.

An additional study of 2737 patients using a remote pharmaceutical care model provided 7758 telepharmacy consultations and resulted in prevention of 1043 adverse drug reactions, affecting 10.4% of patients (3.6 adverse drug reactions per patient). Overall satisfaction was 9.8/10, and mean adherence to the treatment was 95.2% [22].

There is a shortage of health workers, including pharmacists, which poses a challenge for the healthcare system. Telepharmacy, using technology to connect pharmacists and patients, can help alleviate this problem. Telepharmacy means providing pharmacy services when the pharmacist and the patient are not in the same place. Technologies like computers and the internet are used to communicate. Telepharmacy is helpful in areas where there are not enough pharmacists and in areas where it is difficult for people to get to a pharmacy. Researchers looked at articles about telepharmacy published between 2012 and 2018 and found that telepharmacy was helpful in three main ways [23]: supporting clinical services, providing education and managing special types of pharmacies from a distance, and handling prescriptions and making sure people are taking their medicines correctly. Overall, the studies showed that telepharmacy services were effective, and the people who used them were satisfied. However, issues like legal problems make it hard for telepharmacy to become widespread and need to be addressed.

Telemedicine is a complement to face-to-face care, especially in rural areas, which often lack comprehensive medical care due to the sparsity of primary care providers and lack of specialists. Remotely available specialists might also provide consultation to patients in the emergency room or clinic.

The impact of chronic medical conditions on the healthcare system is significant. Telemedicine utilization results in the improvement of chronic medical conditions and a reduction in healthcare costs.

Although there are many potential benefits to telehealth, there are also some potential limitations.

## 8. Potential Limitations of Telehealth

There are some potential limitations, despite many benefits, illustrated in Table 1. One of the main limitations of telemedicine is the limited ability to perform a complete physical exam despite provider-directed self-examination and peripherals. In addition, not all insurers cover telemedicine. With the ongoing pandemic and constant changes in health policy, laws governing the use of telemedicine are constantly changing. Protecting medical data is another concern if the patient accesses telemedicine via a public or unencrypted network. Each state has their own licensing process, limiting clinicians’ ability to practice telemedicine across state lines. Finally, there are the challenges associated with the use of the technology itself, including lack of a strong internet connection, individuals who are not comfortable with the technology, and the technology not working.

## 9. Disparities in Telehealth

Despite increased utilization of telemedicine, differences in health equity and the “digital divide” remain. Older patients, Black patients, and those with Medicaid and Medicare insurance were more likely to have phone-only visits [24]. Black and Latino patients are also less likely than white patients to have high-speed internet [25]. A large single-center study found that non-white patients and those from rural areas used telehealth less in the first month of the COVID-19 health emergency. In the U.S., about 43% of low-income Americans lack fast internet at home [24,26]. To address the lack of infrastructure and technology to engage in telemedicine, the U.S. passed the Infrastructure Investment and Jobs Act in November 2021 to improve internet access and make it a national priority to reduce healthcare differences [27].

## 10. Future Direction of Telehealth

Artificial Intelligence is revolutionizing telemedicine by enhancing diagnostic capabilities and improving healthcare efficiency [28]. Algorithms analyze a large number of data, which helps identify patterns, can predict disease progression, and helps healthcare providers make informed decisions. Patients can interact in real time using chatbots and virtual assistants that are driven by Artificial Intelligence. Providing immediate responses to medical questions and concerns that the patient has leads to timely interventions and positive outcomes in disease management. Artificial Intelligence facilitates remote monitoring of health parameters, helping healthcare providers prevent potential complications and negative outcomes.

Technologies that create an immersive virtual environment and induce a sense of presence, where users perceive, interact, and feel as though they are real, can have a profound impact on telehealth practice and outcomes [29,30].

Virtual reality provides immersive telemedicine experiences, allowing healthcare professionals the ability to conduct remote consultations and surgeries with unprecedented realism. Virtual reality technology enables patients to participate in different medical interventions including educational seminars, virtual therapy sessions, or rehabilitation therapy without the limitation of geographical barriers. Virtual environments for mental health support and therapy can be tailored to the patient’s needs, promoting engagement and improving treatment outcomes.

Augmented reality enhances telemedicine by overlaying digital information onto the real-world environment. Augmented reality can assist healthcare providers in interpreting medical images and can superimpose critical data onto the patient’s real-time view. Surgeons can utilize augmented reality during operations, seeing vital information such as patient history or real-time imaging directly in their field of vision, enhancing precision and reducing errors. The synergy of Artificial Intelligence, virtual reality, and augmented reality in telemedicine holds immense potential for more efficient and comprehensive healthcare.

While the integration of these technologies offers great potential for enhancing and positively impacting the medical field, it is crucial to tackle the challenges related to data security, explore ethical considerations, and address the digital divide to ensure equitable access to emerging healthcare. Robust cybersecurity measures must be in place to protect sensitive patient data, and ethical guidelines should govern the use of Artificial Intelligence in decision-making processes.

The future of telemedicine, shaped by the integration of Artificial Intelligence, virtual reality, and augmented reality, promises a healthcare environment that is more accessible, personalized, and efficient. As these technologies continue to evolve, they hold the potential to positively impact patient care, medical education, and remote healthcare delivery, leading to the betterment of the health of patients. The convergence of telemedicine and advanced technologies represents a significant shift towards a future where healthcare transcends physical boundaries, bringing quality medical services to anyone, anywhere.

## 11. Limitations and Future Directions

This article highlights the breadth of the impact of telehealth within pharmacy and palliative care and provides context for more specific research in the future. Additional systematic reviews are needed to answer specific questions about focused subdomains within telemedicine and palliative care.

## 12. Conclusions

Telemedicine will continue to change the landscape of medicine by augmenting and complementing traditional healthcare. The ease of use of portals, peripherals, universal high-speed internet, and better reimbursement will improve the widespread adoption of telemedicine.

Telemedicine is not a substitute for all face-to-face visits. Telemedicine will continue to grow with advancements in the technology and availability of affordable peripherals to patients at home. The COVID-19 pandemic indeed served as a profound example of how crises can catalyze the quick adoption of new technologies and underscore the challenges associated with staying current with accelerated innovation.

Remote healthcare delivery confers numerous positive outcomes, including improved accessibility, enhanced patient comfort, and efficient care coordination.

The remote monitoring of symptoms facilitates timely adjustments to pain management plans and overall improvement of physical well-being. Telehealth empowers patients and their families to actively participate in care, fostering a supportive environment that extends beyond the confines of a healthcare facility. The transformative impact of telehealth in palliative care eases geographical barriers, enhances patient comfort, and facilitates efficient care coordination. Telemedicine can revolutionize palliative care, providing patients with a more personalized, accessible, and dignified end-of-life experience.

## Figures and Tables

**Figure 1 pharmacy-12-00061-f001:**
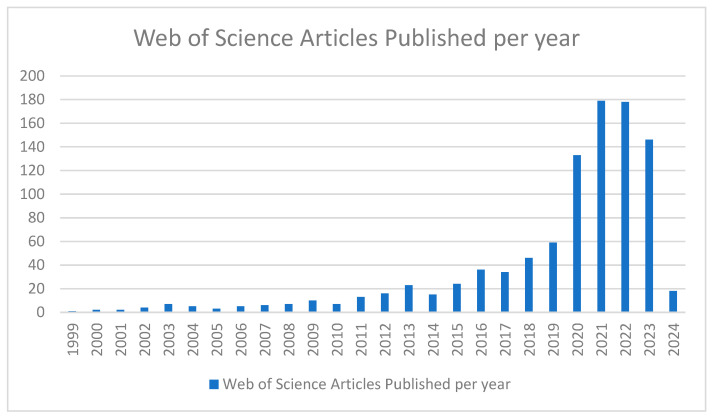
Web of Science Articles published per year with specific search terms.

**Figure 2 pharmacy-12-00061-f002:**
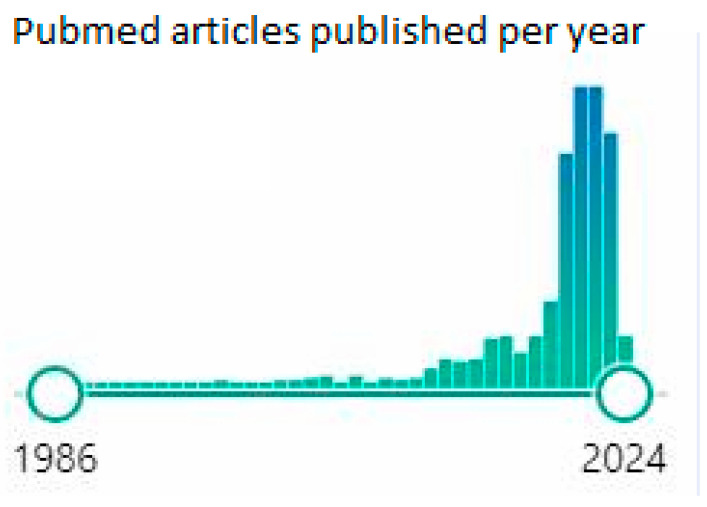
PubMed Articles published per year with specific search terms.

**Figure 3 pharmacy-12-00061-f003:**
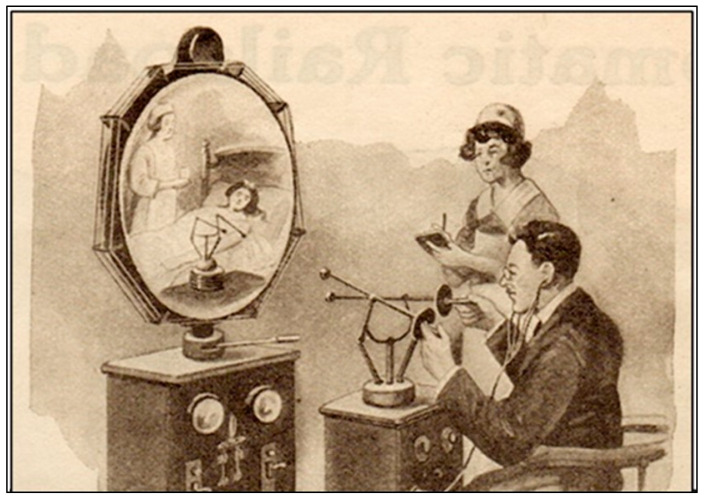
From ‘The Radio Teledactyl’ article by H. Greenback in the February 1925 issue of *Science and Invention*.

**Table 1 pharmacy-12-00061-t001:** Limitations of Telehealth.

Telehealth Limitation and Description
**Insurance Coverage Challenges in Telemedicine** ▪ Not all insurance providers offer coverage for telemedicine; ▪ Regulations related to telemedicine are constantly changing, affecting the availability of insurance coverage and reimbursement.
**Security Concerns for Medical Data Privacy in Telehealth** ▪ Telemedicine sessions accessed on a public or unencrypted network may pose a security risk, jeopardizing patient data.
**Licensing Challenges in Telemedicine Practice** ▪ State laws vary, and clinicians may face restrictions in practicing medicine across state lines.
**Overcoming Technological Barriers in Telehealth** ▪ Weak internet connections can impede the quality of telemedicine sessions; ▪ Patient tech literacy may hinder the effective use of telemedicine.
**Limited Physical Examination in Telemedicine** ▪ Despite provider-directed self-examination and available peripherals, telemedicine has limitations in conducting thorough physical exams.
**Disparities in Telehealth Utilization** ▪ Limited access to telemedicine may amplify existing health disparities among different demographic groups; ▪ Unequal broadband access contributes to disparities in telemedicine utilization; ▪ Disparities in tech literacy affect the ability of certain populations to engage in telemedicine.

## Data Availability

Not applicable.

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
