# Peer review of "Evolution of Telehealth—Its Impact on Palliative Care and Medication Management"

_pharmacy, 2024, doi:10.3390/pharmacy12020061_

Round 1

Reviewer 1 Report

Comments and Suggestions for Authors

Dear Authors

Your work shows an essential view of the use of telehealth in palliative care. It is based on a variety of references.  However,  you only show the positive side of the theme. Only in the conclusion you reserved a few words for the problem.

In my opinion you must perform an integrative review, at least, to present this vital topic nowadays.

Thanks

Hug

Author Response

We thank the reviewers for their time and attention to detail in reviewing our paper. In response to reviewer 1, we have incorporated your kind suggestions and added a table to further clarify and add limitations of telehealth, see table 1 on revised article.

We acknowledge that our review is not a systematic or integrative review but a general review of role of palliative care and the advent of telehealth.  While this article is not a systematic or integrative review, we believe it meets the goal of reviewing different aspects of telehealth including its evolution, current status and future projections.  The role of telemedicine in pharmacy and palliative care is addressed in detail as well as highlighting the potential benefits, challenges and future directions. This article covers multiple areas and for each section a broader view is taken and supported by evidence from published literature. We have endeavored to include a wide range of topics to provide a more comprehensive understanding of how telemedicine impacts pharmacists and the interaction with palliative care, thus a systemic review addressing a specific research question would have limited this ability.

Telehealth Limitation & Description

Insurance Coverage Challenges in Telemedicine

§  Not all insurance providers offer coverage for telemedicine.

§  Regulations related to telemedicine are constantly changing, affecting the availability of insurance coverage and reimbursement.

Security Concerns for Medical Data Privacy in Telehealth

§  Telemedicine sessions accessed on a public or unencrypted network may pose a security risk, jeopardizing patient data.

Licensing Challenges in Telemedicine Practice

§  State laws vary, and clinicians may face restrictions in practicing medicine across state lines.

Overcoming Technological Barriers in Telehealth

§  Weak internet connections can impede the quality of telemedicine sessions.

§  Patient tech literacy may hinder the effective use of telemedicine.

Limited Physical Examination in Telemedicine

§  Despite provider-directed self-examination and available peripherals, telemedicine has limitations in conducting thorough physical exams.

Disparities in Telehealth Utilization

§  Limited access to telemedicine may amplify existing health disparities among different demographic groups.

§  Unequal broadband access contributes to disparities in telemedicine utilization.

§  Disparities in tech literacy affect the ability of certain populations to engage in telemedicine.

Reviewer 2 Report

Comments and Suggestions for Authors

The introduction effectively outlines the role of palliative care and the advent of telehealth, providing a comprehensive overview.

    • Aspect to Improve: To enhance the introduction, it might be beneficial to integrate more specific examples or case studies that underscore the unique challenges and opportunities telehealth presents in palliative care.

The references seem pertinent and comprehensive, enhancing the scholarly depth and context of the paper. However, as a review, I have several concerns about the study.

While the paper aims to review the evolution of telehealth in palliative care, the approach does not appear to be systematically structured. There's a need for a more clearly defined research methodology.

    • Aspect to Improve: The paper should explicitly state the databases searched, the keywords used, the time frame of literature considered, and the inclusion and exclusion criteria for selecting articles. This would add rigour and reproducibility to the research.

The manuscript lacks a clear description of the methods used for gathering and selecting literature, which is a critical component of a review article.

    • Aspect to Improve: The authors should detail the systematic approach taken, including search strategies, selection criteria, and analysis methods. This is what type of review? Just a search on the Web, but where? According to which criteria? The authors use a scoping, systematic method? Following which guideline?

While the paper attempts to synthesize information from various sources, the lack of a systematic approach to literature selection and analysis may lead to a biased or incomplete overview.

The discussion and conclusions are based on the presented literature, but the lack of a systematic approach raises questions about the comprehensiveness and bias of the sources selected.

    • Aspect to Improve: The paper should address the limitations of the unsystematic approach and consider the implications for the strength and validity of the conclusions drawn.

In summary, while the manuscript aims to contribute to the academic discussion on telehealth in palliative care, the lack of a systematic approach in its literature review significantly undermines its methodological quality. To improve, the authors should provide a clear and detailed description of their research methodology, ensuring the paper meets the scholarly standards expected for a review article. This includes defining the scope, databases searched, search terms used, and article selection and analysis criteria. The manuscript can significantly enhance its credibility, rigour, and overall contribution to the field by addressing these concerns.

If not, this paper is more like a concept paper than a review. A concept paper typically presents an idea, theory, or perspective, often to propose research or provide a broad overview of a topic. However, even as a concept paper, providing a transparent methodology or approach to how the information was gathered, synthesized, and analyzed is crucial. This could include a discussion of the conceptual framework, the perspective or lens through which the topic is viewed, and the rationale for including certain types of information or sources. Transparency in the approach, even if it is not a systematic review, adds credibility and allows readers to understand the basis for the conclusions drawn.

Author Response

We acknowledge that our review is not an integrative review but a general review of role of palliative care and the advent of telehealth.  While this article is not an integrative review, we believe it meets the goal of reviewing different aspects of telehealth including its evolution, current status and future projections.  The role of telemedicine in pharmacy and palliative care is addressed in detail as well as highlighting the potential benefits, challenges and future directions. This article covers multiple areas and for each section a broader view is taken and supported by evidence from published literature. We have endeavored to include a wide range of topics to provide a more comprehensive understanding of how telemedicine impacts pharmacists and the interaction with palliative care, thus a systemic review addressing a specific research question would have limited this ability.

Reviewer 3 Report

Comments and Suggestions for Authors

Dear authors, the article is interesting although the topic is very, very covered. Undoubtedly the most interesting part is that of artificial intelligence. You did well to include this part but I think we can write an article just on these aspects. Another interesting part concerns the limits to telemedicine and telepharmacy activities in palliative care. In my opinion, two paragraphs should be inserted in the limits part. The first concerns the patient and understanding informed consent to palliative oncology therapy online. Or you collect the consent when you come to the hospital, in short, insert something about the acquisition of informed consent.

The second paragraph, again in the limitations, concerns the malfunction of the digital connection systems used to contact the patient. That is, what happens if the electronic equipment does not work and harm is caused to the patient? Whose responsibility is it? I think that by adding these two small paragraphs in the limitations the article is complete and can improve a lot.

Author Response

We thank the reviewers for their time and attention to detail in reviewing our paper.  In response to reviewer 3, we have incorporated your kind suggestions and added a table to further clarify and add limitations of telehealth. 

Telehealth Limitation & Description

Insurance Coverage Challenges in Telemedicine

§  Not all insurance providers offer coverage for telemedicine.

§  Regulations related to telemedicine are constantly changing, affecting the availability of insurance coverage and reimbursement.

Security Concerns for Medical Data Privacy in Telehealth

§  Telemedicine sessions accessed on a public or unencrypted network may pose a security risk, jeopardizing patient data.

Licensing Challenges in Telemedicine Practice

§  State laws vary, and clinicians may face restrictions in practicing medicine across state lines.

Overcoming Technological Barriers in Telehealth

§  Weak internet connections can impede the quality of telemedicine sessions.

§  Patient tech literacy may hinder the effective use of telemedicine.

Limited Physical Examination in Telemedicine

§  Despite provider-directed self-examination and available peripherals, telemedicine has limitations in conducting thorough physical exams.

Disparities in Telehealth Utilization

§  Limited access to telemedicine may amplify existing health disparities among different demographic groups.

§  Unequal broadband access contributes to disparities in telemedicine utilization.

§  Disparities in tech literacy affect the ability of certain populations to engage in telemedicine.

Round 2

Reviewer 1 Report

Comments and Suggestions for Authors

Dear Authors

I realised that you did not accept my suggestions. So my decision is the same.

Thanks

Kind regards

Author Response

We thank the reviewers for their time in reviewing our manuscript and attention to detail. We have revised our manuscript taking all their excellent points into consideration. 

  1. Reviewer Suggestion: Your work shows an essential view of the use of telehealth in palliative care. It is based on a variety of references. However, you only show the positive side of the theme. Only in the conclusion you reserved a few words for the problem.

In response to reviewer 1 Suggestion 1:  we have incorporated your kind suggestions and added a table to further clarify and add limitations of telehealth, see table 1 on the revised article.

  1. Reviewer 1 Suggestion 2: In my opinion you must perform an integrative review, at least, to present this vital topic nowadays.

In response to reviewer 1 Suggestion 2: We acknowledge that our review is not a systematic or integrative review but a general review of the role of palliative care and the advent of telehealth.  While this article is not a systematic or integrative review, we believe it meets the goal of reviewing different aspects of telehealth including its evolution, status, and future projections.  The role of telemedicine in pharmacy and palliative care is addressed in detail as well as highlighting the potential benefits, challenges, and future directions. This article covers multiple areas and for each section, a broader view is taken and supported by evidence from published literature. We have endeavored to include a wide range of topics to provide a more comprehensive understanding of how telemedicine impacts pharmacists and the interaction with palliative care, thus a systemic review addressing a specific research question would have limited this ability.

Reviewer 2 Report

Comments and Suggestions for Authors

Thank you for the revisions made to your manuscript on the evolution of telehealth in palliative care and its implications for medication management. Including case studies and more specific examples, such as the integration of telehealth in Mr. Alex's palliative care plan, significantly enriches the narrative, providing concrete illustrations of telehealth's potential benefits.

However, in alignment with the feedback from the first review round, there are still areas needing further refinement to enhance the manuscript's rigour and comprehensiveness:

  1. Research Methodology: The addition detailing the general review approach is acknowledged. Yet, it would be beneficial to further specify the scope of the literature reviewed for enhanced clarity and academic rigour. This could include detailing the years considered, databases searched, and keywords used. Even if a systematic review methodology were not employed, providing this information would bolster the paper's credibility.
  2. Systematic Approach: While the manuscript aims to offer a broad overview, incorporating elements of a more structured review could mitigate potential biases. For instance, mentioning any efforts to ensure a wide representation of perspectives within the telehealth domain would be valuable.
  3. Limitations and Future Directions: Acknowledging the review's general nature is appreciated. Expanding on this by explicitly stating the limitations of such an approach and suggesting specific areas for future systematic reviews could offer readers a clearer understanding of the context and potential research gaps.

The manuscript contributes to understanding telehealth's role in palliative care. These suggestions could further enhance its impact, providing a robust foundation for future research and implementation strategies in this rapidly evolving field.

Author Response

Response to review: Evolution of Telehealth – Its Impact on Palliative Care and Medication Management

We thank the reviewers for their time in reviewing our manuscript and attention to detail. We have revised our manuscript taking all their excellent points into consideration. 

  1. Research Methodology: The addition detailing the general review approach is acknowledged. Yet, it would be beneficial to further specify the scope of the literature reviewed for enhanced clarity and academic rigor. This could include detailing the years considered, databases searched, and keywords used. Even if a systematic review methodology were not employed, providing this information would bolster the paper's credibility.

In order to enhance clarity of the scope and intention of this unstructured review we have added a paragraph (‘Article Scope and Intention’) documenting the keywords used during our search of both Web of Science and Pubmed, years considered, and the most recent date that both sites were accessed. 

  1. Systematic Approach: While the manuscript aims to offer a broad overview, incorporating elements of a more structured review could mitigate potential biases. For instance, mentioning any efforts to ensure a wide representation of perspectives within the telehealth domain would be valuable.

We regret not having made clearer our intention and thus have added a paragraph documenting the wide representation of perspectives we endeavored to include:

“This broad overview includes a wide range of topics and perspectives within the telehealth domain; drawing on experience of telemedicine within chronic health conditions like: type 2 diabetes, dyslipidemia, hypertension/cardiovascular medicine, depression, cancer pain and care during the covid-19 pandemic, as well as within the field of Palliative Care.  We explore various models of care and geographic areas impacted by telehealth, and how telemedicine impacts patients, caregivers, medical professionals and health care systems, including potential benefits, challenges and future directions.”

  1. Limitations and Future Directions: Acknowledging the review's general nature is appreciated. Expanding on this by explicitly stating the limitations of such an approach and suggesting specific areas for future systematic reviews could offer readers a clearer understanding of the context and potential research gaps.

The authors thank the reviewer for this excellent suggestion and have added a paragraph entitled ‘Limitations and Future Directions’ with the hopes that this article provides a framework for additional papers in the future.

Sincerely,

Dr Syed Imam

Round 3

Reviewer 1 Report

Comments and Suggestions for Authors

Dear authors

I appreciated your reply to my concerns and questions.

The most important changes remain in this version. However, because the journal accepts this type of publication, I will recommend "to accept."

Kind regards

Reviewer 2 Report

Comments and Suggestions for Authors

The revisions made have notably enhanced the manuscript's methodological quality, credibility, and academic rigor. The inclusion of a specific case study, clarification of the research methodology, acknowledgment of limitations, and identification of future research directions are commendable improvements. Therefore, I recommend that the manuscript be accepted.